# The Potential Impact of COVID-19 on the Chinese GDP, Trade, and Economy

**Zohal Habibi [1], Hamed Habibi [1,\*] and Mohammad Aqa Mohammadi [2,\*]**

[1] Department of International Economics, School of Economic, Hefei University of Technology, Hefei 230009, China; zohalhabibi20@gmail.com

[2] College of Horticulture, Fujian Agriculture and Forestry University, Fuzhou 350002, China

\* Correspondence: hamed.habibi1@gmail.com (H.H.); 1161906001@m.fafu.edu.cn (M.A.M.); Tel.: +86-139-6672-3052 (H.H.); +86-180-2088-3292 (M.A.M.)

**Abstract:** COVID-19, a novel Coronavirus SARS-CoV-2, has wreaked havoc on global financial markets, economies, and societies. For example, this study looks at the impact of COVID-19 on the Chinese economy and its policy responses (fiscal, monetary, and institutional). This study also examines future issues. This study is timely and essential for policymakers and investors worldwide because of China's size, contribution to global growth, and growing influence. The research shows that the presence of COVID-19 in China has global implications. Because of the virus threat, foreigners avoid mixing with the Chinese. Global tourists have cancelled their plans to visit China, and Chinese tourists cannot visit foreign countries. The rapid spread of the COVID-19 in China has halted normal life. The intensification of the COVID-19 may have long-term effects on China's economy.

**Keywords:** COVID-19; trade; GDP; economy

## 1. Introduction

The outbreak of the COVID-19 pandemic affected the global economy. COVID is an enormous group of infections that can cause several illnesses, such as minor cough, extreme respiratory severe condition (SARS), and Middle East respiratory disorder (MERS). In China, a worldwide spread of the virus COVID-19 has broken out, as it is the developed form of Coronavirus (Clinics 2020) and numerous other COVID strains contracted from bat populaces. The symptoms of this virus appear after 2to 14days in patients, and most patients face breathing problems, fever, and sickness. The seriousness of the indications of the new COVID-19 infection shifts from individual to individual very quickly. This infection is perceived inside and out, yet most seriously sick patients are older or are experiencing other genuine and persistent illnesses. This virus is more deadly for patients with respiratory illnesses, such as influenza or other illnesses (Clinics 2020).

The first case was registered in December 2019. It is considered a deadly virus that infected more than 52 million people globally. The death rate was also high, as 1.2 million people died from this fatal virus by November 2020 (Kuckertz et al. 2020). So, it has a broad negative impact on the lives and health of people. Regardless of the health sector, COVID-19 negatively impacted economic activities. The COVID-19 pandemic badly affected almost all sectors of the economy by reducing their performance and activities. Hoteling, transportation, manufacturing, tourism, and trade industries experienced a remarkable decline in their revenues.

Some statistics show that COVID-19 is negatively impactful for global economies, such as that the world's GDP declined 2% below its benchmark, and developing countries declined their GDP by about 2.5%. Industrial and developed countries reduced their GDP by about 1.8% (Islam et al. 2020). This virus spread worldwide very quickly, and most countries closed their borders to the entire world. This caused a considerable reduction in international trade, immigration, and tourism. TheWorld Health Organization(WHO)

suggested these isolation policies, so developing countries also employed these policies. The services sector suffered greatly from this pandemic because all countries restricted social interaction. Therefore, many labour forces became unemployed, especially those engaged in the tourism and transportation sector. The manufacturing industry was also shut down, and it caused unemployment. The demand and supply mechanisms were negatively disturbed by this situation (Kuckertz et al. 2020).

The COVID-19 virus commonly spreads through sniffles, touch, or talks that create tiny drops in the environment. These beads are too hefty to ever stay in the air and drop on the floor or onto different surfaces. Suppose that people are inside and one meter away from a COVID-19 patient. In that case, they might be tainted by breathing in the infection or touching a polluted surface before washing hands and afterward touching their eyes, nose, or mouth (WHO 2020). Currently, no antibody or any other antiviral medication is available that can forestall or treat the COVID-19 infection. However, tainted individuals should get treatment and maintain social distance, while severe patients should be hospitalized. Most patients recover in the wake of accepting a steady treatment. Immunizations and particular medication treatments are being investigated and tried through clinical preliminaries. The World Health Organization (WHO) is, until further notice, planning antibody and medication advancement to forestall and treat COVID-19 (WHO 2020). As per the most recent forecast model of Harvard University, the pandemic may have a more genuine episode, and there may, in any case, be a repeat before 2025. Considering the information demonstrated, the analysts proposed to foresee that without antibodies or other powerful medicines, the pandemic may even repeat before 2025. Accordingly, the investigation highlighted a critical chance: this pandemic, which has given many nations bad dreams, will have the danger of returning what is to come (Yu et al. 2021).

Similarly, given the information displayed, the analysts propose that without antibodies or other successful medicines, COVID-19 may even repeat before 2025, according to a recent statistics review (Qiu 2021). For group invulnerability, it is necessitated that adequate treatment or precautions be tainted. The absence of accessible hosts stops the transmission of the infection, and it occurs after far and wide openness. With an illness as pervasive s COVID-19, specialists accept that more than 66% of the populace would be resistant so that crowd insusceptibility can be made. Therefore, much work is needed to save humanity from COVID-19 (Craven et al. 2020).

The new coronavirus (COVID-19) outbreak has resulted in a public health catastrophe and an economic crisis on a global scale, affecting a wide range of sectors. Having been initially hit by the COVID-19 pandemic, China was also the first country to recover and resume economic activity. COVID-19′s impact on SMEs in China was investigated in February 2020, and this report proposes governmental initiatives to mitigate the detrimental consequences of the pandemic on small- and medium-sized enterprises (SMEs) (Lu et al. 2021).

There are many fiscal policies, such as subsidies, increases in spending, loans, and equity purchases. Chinese governments chose to extend long-term loans to the industrial sector. These long-term loans will help the industrial and business sectors continue their business. Moreover, it will ensure the employment of the labour class and the provision of commodities to households. Long-term loans will build business people's confidence, and they will adequately run the business because they have no fear of repaying loans early (Meng et al. 2020). This loan amount will promote economic activity and help other linked industries. However, these loans should provide a low-interest rate compared to the market rate, and governments may advise the banks about the timely and easy provision of credit. So, government loans are senior to other banks loans. The subsidy and transfer payment have many disadvantages, such as it will not precisely promote economic activity. The free-of-cost government assistance may make the people in business passive and will not work to promote economic activities.

*Objectives of the Study*

The study looks at the impact of COVID-19 on the Economy of China, and this is the study general objective. The study uses the first wave of COVID-19 for empirical estimation. Different indicators measure economic performance, but this study uses GDP growth, industrial sector, and international trade. The study is followed by Liu, 2021 on how the Chinese economy was affected by Covid-19. There are a set of specific objectives by which general-purpose can be achieved.

- To explore the relationship between COVID-19 and GDP growth.
- To explore the relationship between COVID-19 and the industrial sector.
- Impact of COVID-19 on international trade of China.
- To deliver the theoretical context of the COVID-19 pandemic.
- To provide the trend analysis of COVID-19 and the economic growth of China.
- Suggest policy implications so that the negative impacts of COVID-19 can be minimized.

## 2. Review of Previous Studies Related to COVID-19

Many studies discussed this pandemic; (Dhar 2020) tried to explore the impact of COVID-19 on the economy of China. The study found the effects of COVID-19 on GDP, the balance of trade, and the stock exchange. COVID-19 negatively impacted Chinese exports and imports. Exports declined up to 17.5%, and imports decreased by 4%. The outbreak of COVID-19 also caused the decline of share prices, such as SSECI, which decreased by 36 points. Social distancing and isolation policies also minimized production and other economic activities. So, the GDP of China did achieve its targeted value due to the spread of the COVID-19 pandemic.

Liu and Hu (2020) examined the impact of COVID-19 on the Chinese economy. The study used the neoclassical growth model. The study found that while the COVID outbreak did not affect China's domestic demand, the virus's global spread reduced the Chinese goods market. With time, China's social distancing policies caused a significant decline in production. The global spread of COVID-19 and the WHO-recommended border closures harmed trade balance. The study could not examine the long-term effects of COVID-19 on the Chinese economy. So, as COVID-19 spread globally, the saving rate increased in China, which may be used in economic activities.

The OECD (Yang and Deng 2021) studied the impact of COVID-19 on the global and Chinese economies. The study used the NIGEM macro model and suggested an exogenous fiscal policy to mitigate the study's adverse effects. In the worst-case scenario of COVID-19, more spending, lower taxes, and subsidies were beneficial. According to the study, China's demand fell by 2% to 4%. Commodity prices fell 10% globally, and the global GDP fell 0.5% in 2020. Due to the global spread of COVID-19, China's GDP fell 0.2%, and imports fell 6.0%.

To investigate the impact of COVID-19 on the economy (Wei et al. 2021), the study found that the outbreak of COVID-19 disrupted production, business, and households' standard of living. As the industrial sector is vital to an economy, COVID-19 harms their industry, causing many businesses to close. This situation makes it difficult for businesses to manage credit, staff, and expenses.

Luo et al. (2020) studied the Chinese economy and found several impacts of COVID-19. Globalization has accelerated the spread of COVID-19, which began in China. The study found that China's social distancing and isolation policies slowed the spread of COVID-19 but hampered economic activity. China's declining production and border closures affected global and Chinese economies. The study found a 3% global GDP decline in 2020, with developing countries losing 4% to 7%. The author used graphs to examine China's GDP growth and fluctuations in other economic sectors.

Wang and Su (2020) studied COVID-19's economic effects in China. The author estimated that COVID-19 broke out, and high risks cities shut down in January 2020. The study also investigated how China could spread COVID-19, which had no new cases in

March. The study found that the COVID-19 outbreak reduced consumption. Shutdown policies also harmed other economic sectors, such as industry, transportation, tourism, and education. Moreover, the shutdown and isolation policies reduced investment, resulting in lower GDP growth. Export and import levels fell due to WHO restrictions, the author added.

COVID-19 had a negative impact on all sectors of the economy, but this study (Zhang et al. 2020) focused on the agriculture sector of China. This study also investigates the impact of COVID-19 on macroeconomic indicators. The author used SAM multiplier analysis to discover many results. The study found that China's GDP fell 6.8% in the first quarter of 2020, with the agriculture sector losing RMB 0.26 trillion. This represents a 7% loss in agricultural value-added and a 27% loss in agricultural employment. In 2020, the agricultural sectors target growth rate was 1.1%, but it only reached 0.4%. The author stated that the agricultural sector suffered due to the drop in global demand for Chinese agricultural products.

COVID-19 and China's agricultural exports were examined by Lin and Zhang (2020). The COVID-19 outbreak disrupted agricultural supplies from China to the world. The study focused on the agriculturally rich Fujian Province in China. This study used regression analysis and found some critical results. With the spread of COVID-19 came a decline in agricultural exports, which worsened as international borders closed. The level of medical herb exports correlated negatively with the COVID-19 pandemic.

COVID-19 also harmed edible fungus, horticultural, oilseed, and edible oil products. Due to the transportation shutdown, the labour force could not migrate to other sectors. The authors propose that the government should aid and subsidize this sector.

Lu et al. (2020) studied China's social policies in the presence of COVID-19. COVID-19 had many adverse effects on China's economy, such as slowing industrial growth, decreasing GDP, reducing national exports and imports, and increasing unemployment. The author explored how the Chinese government provides financial assistance, social insurance, and social welfare. People receive social, medical, pension, and unemployment benefits. In this COVID-19pandemic, special medical, educational, and legal aid was available. The author also stated that China's government provided special assistance to children and disabled people.

### 2.1. COVID-19 and Chinese Economy

The economic condition and its involvement in international trade make China important for all other countries. The financial or manufacturing fluctuations in China can disturb the trade balance globally. The current situation of COVID-19 hit the Chinese economy badly, and it also negatively impacted other economies around the world. Manufacturing activities stopped when COVID-19 spreads around the countries. The social distancing rules cause the underutilization of labour and capital in China, increasing the cost of production. Moreover, shutting down international borders and other transportation restrictions made the exports difficult for China, and it suffered huge losses, such as a reduction in its exports by 3.7% (Liu and Hu 2020).

Figure 1 presents the export trends of the Chinese economy from 2017 to December 2020. The figure shows that exports of China remarkably declined in January 2019 due to the worst period of the COVID-19 pandemic. Then the passage of time and development of precautions against COVID-19 made a positive impact on the exports of China. By the end of 2020, exports were at their highest level.

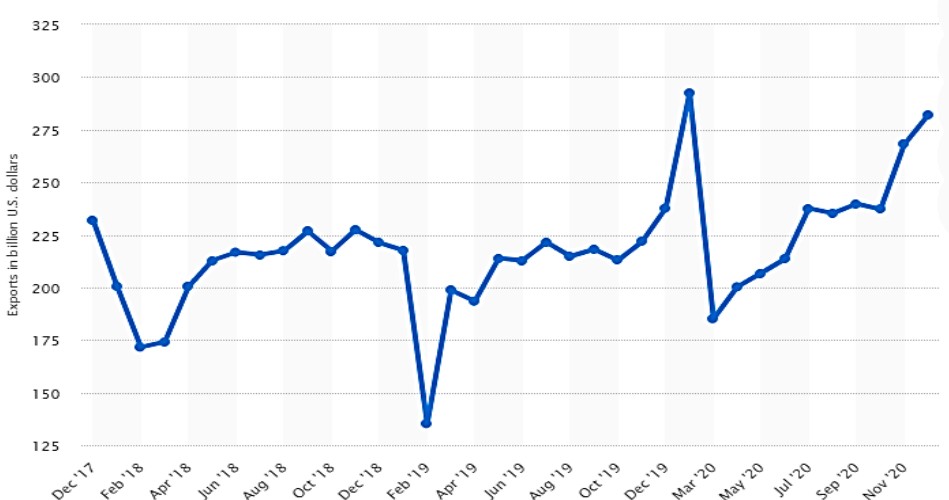

**Figure 1.** Exports of China. Source: The figure reproduced with permission from (National Bureau of Statistics of China 2021).

It is also seen that during the worst time of the pandemic, there was a remarkable decline in the investment sector, and extensive stock of the automobile industry remains unsold. The outbreak of COVID-19 caused the reduction of foreign direct investment, tourism, and other business trips. COVID-19 has the property of quick spread among people, so the government of China strictly followed the advisory of social distancing (Wong et al. 2020). Further preventions were taken, such as closing educational activities, private and government businesses, non-governmental organizations, and international trade. However, necessities and lifesaving commodities can transfer at the international level. The only objective of these measures is to minimize the spread of COVID-19, and China achieved this objective by these measures (Liu et al. 2020). Before COVID-19, the world faced many other types of pandemics and suffered the loss of massive deaths (Keogh-Brown et al. 2020). However, the COVID-19 spread quickly and covered the entire world. Many researchers, such as Allen et al. (2008), suggested that an increase in globalization and more trade of animals always have a more significant possibility of transmitting diseases. However, Keogh-Brown et al. (2020) explored the negative impacts of COVID-19 and suggested that the current pandemic is the most harmful and deadly. As COVID-19 compared to the previous pandemics and the number of death given in Table 1. Many studies used different variables to measure COVID-19, such as total cases of COVID-19, death rate, and total patients that recovered from COVID-19 (Alfani and Murphy 2017).

**Table 1.** Pandemic a historical perspective.

| Pandemic Name | Number of Deaths | Time Duration |
| --- | --- | --- |
| Black Death | 75,000,000 | 1331 to 1351 |
| Plague of Italy | 281,000 | 1623 to 1632 |
| Plague of Seville | 2,000,000 | 1647 to 1652 |
| London great Plague | 100,000 | 1665 to 1666 |
| Marseille Plague | 110,000 | 1720 to 1722 |
| Cholera | 100,000 | 1816 to 1826 |
| Cholera (2nd Pandemic) | 100,000 | 1829 to 1851 |
| Cholera in Russia | 1,000,000 | 1852 to 1860 |
| Flue Pandemic worldwide | 1,000,000 | 1889 to 1890 |

**Table 1.** *Cont.*

| Pandemic Name | Number of Deaths | Time Duration |
|---|---|---|
| Cholera (6th Pandemic) | 800,000 | 1899 to 1923 |
| Pandemic of Encephalitis Lethargica | 1,500,000 | 1915 to 1926 |
| Flu of Spanish | 100,000,000 | 1918 to 1920 |
| Asian Flue | 2,000,000 | 1957 to 1958 |
| H1N1 pandemic | 205,000 | 2009 to 2010 |

Source: Plague and lethal epidemics in the pre-industrial world.

*2.2. Short-Term and Long-Term Impacts of COVID-19 and the Chinese Economy*

The epidemic of COVID-19 made many short-term and long-term shocks, and there are many types of research, such as Chen et al. (2020) and Liu et al. (2020), exploring the COVID-19 impacts and government policy of this pandemic. In a short-term analysis, the Chinese government took many emergency steps in response to COVID-19. The isolation and social distancing policies negatively impacted the domestic demand for commodities, such as decreased domestic demand. Moreover, the global market of Chinese products also decreases due to isolation and social distance policies. The categories of stages by which we can divide the impact of COVID-19 are as follows.

The first stage substantially negatively impacted consumption because it was a spring festival. All type of shopping malls was closed during this period, so there was a remarkable reduction in the consumption of commodities. Moreover, tourism, industrial, and retail activities were minimized, negatively impacting the country's consumption level. The restaurant and hoteling sectors suffered a lot, and about 93% of catering companies closed their business due to low demand for their services and goods.

The second stage comes right after the spring festival, and workers have to return to their careers, but due to COVID-19, they were unable to perform their duties. This caused the reduction of the production of commodities; moreover, less availability of transportation makes the transfer of the labour force impossible. So, the industrial sector suffered a lot from social distancing initiatives. The statistics indicate that about 50 million workers could not travel to their work due to isolation policies. Thus, this created unemployment and diminished the standard of living in China (Liu and Hu 2020).

The worldwide spread of COVID-19 referred to the third stage, and the demand for Chinese products reduced at the global level. Developed and developing nations strictly followed the policies and precautions suggested by the WHO about isolation. So, worldwide policies of isolation and social distancing put the borders close, then caused a substantial negative impact on Chinese international trade. The blowout of COVID-19 in the USA and European countries caused a considerable decline in the demand for Chinese products (Shen et al. 2020).

In the early months of 2020, such as January and February, China's macroeconomic information indicated negative development since measurements became accessible. Among these statistics, value-added to the modern industrial sector diminished by 13.5%. The services sector recorded a reduction of 13.0%, and absolute retail deals of social customer merchandise decreased by 20.5%. To forestall and stop COVID-19, China has paid a hefty monetary cost.

Figure 2 presents the industrial production growth trends of the Chinese economy from 2017 onward. The figure shows that industrial production of China remarkably declined after mid-2019 due to the worldwide spread of COVID-19. The demand for Chinese products declined mainly due to social distancing and the lockdown of international borders. Then the passage of time and development of precautions against COVID-19 positively impacted the industrial sector of China. By the end of 2020, industrial sector performance was at its highest level.

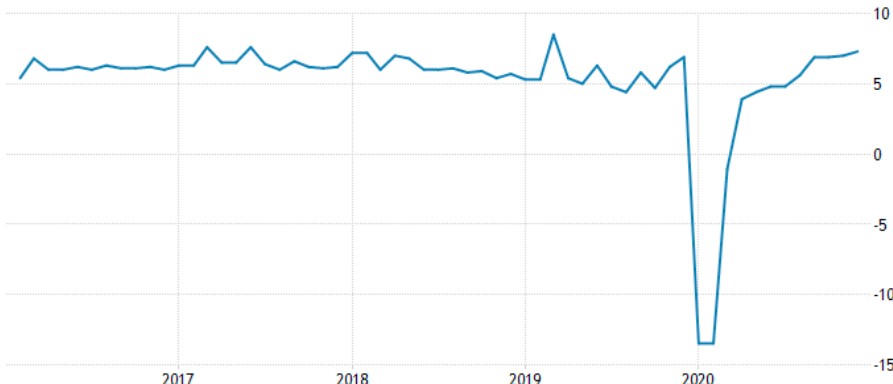

**Figure 2.** Industrial production of China. Source: The figure reproduced with permission from (National Bureau of Statistics of China 2021).

After the flare-up of COVID-19, China immediately dispatched a bundle of plans. To guarantee "hostile to plague", the general public effectively gave cash and materials to guarantee "hostile to plague," and the Chinese government immediately organized particular reserves. Simultaneously, the Chinese government has additionally declared different approaches, including financial arrangements, charge strategies, money-related arrangements, modern strategies, and business strategies. Regarding monetary interpretation, a blend of duty decrease, expense decrease, and endowments was given, and "Against Epidemic Thematic Bond" was given. The government has given particular treatment to businesses influenced by the plague, e.g., transportation, catering, travel, convenience, expedited service, standard avionics, and different companies. As indicated by fundamental gauges, the overall population spending shortage rate in 2020 may increase from 2.8% of GDP to about 3%.

Figure 3 explores the nominal fixed investment trends for the Chinese economy between the years 2019 and2020. The figure shows that investment inChina remarkably declined due to the outbreak of COVID-19. Then the passage of time and development of precautions against COVID-19 had a positive impact on the investments of China. By the end of 2020, it had a negative growth rate, and COVID-19 also had a low growth rate.

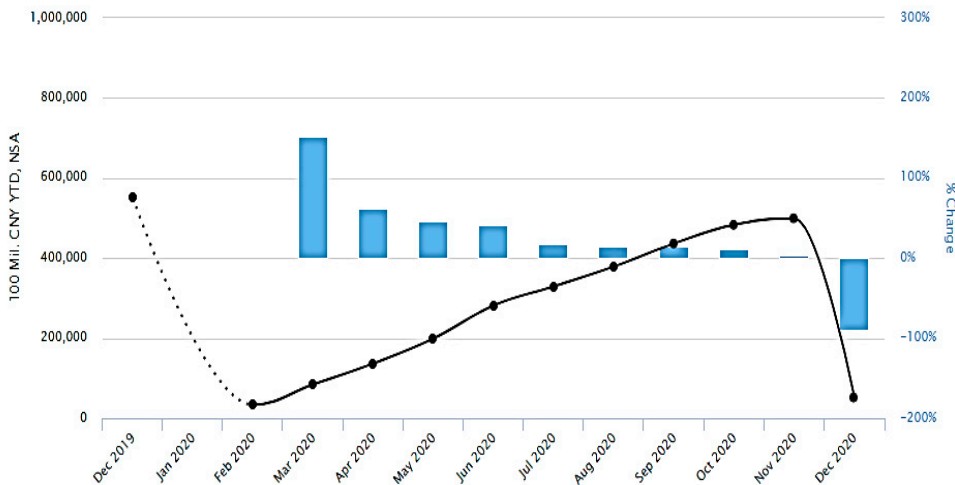

**Figure 3.** Nominal Fixed Investments.Source: The figure reproduced with permission from (National Bureau of Statistics of China 2021).

Regarding the financial approach, it has additionally kept up enough for free. Through measures like balance sheet expansion, reserve requirement ratio RRR-cutting, and strategy loan cost-cutting, the People's Bank of China (POBC)guarantees adequate liquidity and no expansion in financing costs for the real economy, simultaneously as financial strength.

As of now, the focal banks unique enemy of scourge renegotiating portion has arrived at 800 billion Yuan. Simultaneously, under the state of impeded outside interest and the level foreign trade holds, PBOC focused on RRR–slicing to help develop all-out social financing. A more straightforward improvement plan comes from a "new framework." In a request to battle the negative effects of COVID-19, as of in 1 March 2020, 13 territories, including Beijing, have delivered venture plans for critical activities in 2020 with up to 33.8 trillion Yuan. Among them, "new foundation" is exceptionally anticipated. "New framework" is unique concerning the customary foundation, for example, railroad, parkway, air terminal, and primarily incorporates seven perspectives: "5G foundation, UHV, intercity rapid rail line and metropolitan rail travel, new energy vehicle charging heaps, huge server farms, man-made consciousness, mechanical Internet" (Chen et al. 2020).

Figure 4 presents the trade balance of the Chinese economy from 2017 to quarter 2 of 2020. Figure 4 shows that China's trade balance declined with the outbreak of COVID-19, such as in 2019, due to the severe conditions of COVID-19. Then, the passage of time and the development of precautions against COVID-19 positively impacted China's trade balance. By the end of 2020, the trade balance had a positive figure, showing that the implications of COVID-19 were minimized.

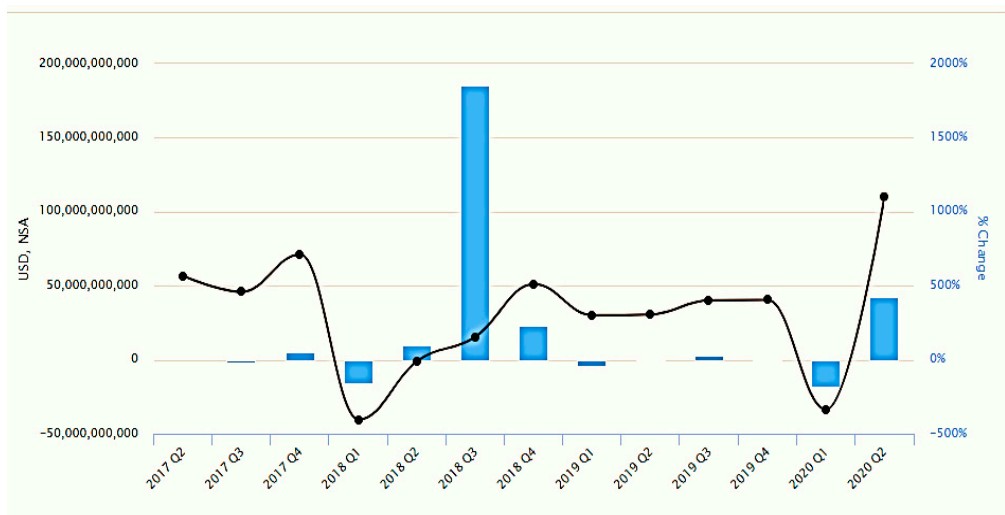

**Figure 4.** Current account balance. Source: National Bureau of Statistics of China The figure reproduced with permission from (National Bureau of Statistics of China 2021).

Regarding business strategy, the public authority decreased the trouble on endeavours through the expense and charge decrease arrangements from one perspective. Then again, the public administration is effectively extending business channels. These channels incorporate developing enrollment plans for state-possessed undertakings, growing business at the grassroots level (for example, supporting agribusiness, instruction, clinical consideration, and destitution lightening, and so on), extending the size of business learners, sorting out "Cloud" job fairs for graduates, and properly postponing acknowledgment. It tends to be seen that under the extreme effects of the plague, China has consistently embraced general wellbeing approaches, financial arrangements, money-related strategies, mechanical approaches, and business arrangements, which are exceptionally focused on and very ground-breaking. Consequently, this paper will generally accept that the prior quarter ought to be the lower part of the transient economy. Afterward, it will show a "base bounce back what is more, generally adequacy". Consistently, the probability of finishing the business and development targets set by the Central Economic Work Conference stays high. What is more, on the planet, the more noteworthy recuperation of China's economy is expected to lead to the worldwide economy's recovery (Zhao et al. 2020).

## 3. Materials and Method

The study's primary purpose is to show the impact of COVID-19 on China's economic growth. We obtained monthly data from China from January 2019 to December 2020, and the data is obtained from WHO, World Development Indicators (WDI), and the Chinese economy.

### 3.1. Econometric Estimation

Different econometric techniques are used for estimation of analyses. These stepwise econometric techniques are detailed below:

Unit Root Analyses

Different unit root tests check the model of the variables for China to detect integration on both variables. The trial of Levin et al. (2002) is begun by way of the Augmented Dickey Fuller (ADF) Equations (1) and (2) that engages the ordinary procedure of the panel unit root survival through the payment of conflicting lag order integration in cross-sections of the panel. The null hypothesis (H0) indicates the non-stationarity, whereas the Alternate Hypothesis (H1) indicates the stationary series.

$$\Delta y_{it} = \alpha_0 y_{it-1} + \sum_{p=1}^{ni} a_{1ip} \Delta y_{it-p} + \lambda_{it} + \varepsilon_{it} \tag{1}$$

$$a_0 = n - 1$$

Im et al. (2003) a panel unit identifies the ADF of each class of panel assigned to a particular forum by measuring the average t-statistics of ADF statistics $\bar{t}_{NT}$ with a zero interval in the above Equation (1) of the ADF, it has been pointed out that Im et al. (2003) indicates the critical values stated in each class (*n*) and the length of the series and for the variables that are implanted by constant or by constant with trend to each other. For non-zero intervals, it indicates that the normal distribution is given in the Equation (2) below Im et al. (2003).

$$X_{\bar{t}_{NT}} = \frac{\sqrt{P}\left[\bar{t}_{NT} - N^{-1}\sum_{i-1}^{N} F\left(\bar{t}_{NT}(N_i)\right)\right]}{N^{-1}\sum_{i-1}^{N} \text{var}\left(\bar{t}_{it}(N_i)\right)} \tag{2}$$

where is the mean; $\text{var}\left(\bar{t}_{it}(N_i)\right)$ is the variances of the ADF regression of t-statistic $F\left(\bar{t}_{NT}(N_i)\right)$ caused to be by Im et al. (2003), with reverence of various lags, series lengths, and the assumptions that support different test equations.

### 3.2. Model Specification

In recent decades, the Chinese economy has been tormented with air contamination (Zhang et al. 2020) because of massive industrialization and anthropogenic exercises. To this end, the current investigation endeavours to approve an immediate connection between financial development and fossil fuel byproducts in China somewhere in the range of 2019 and 2020, to set up the flexibility connection between fossil fuel byproducts and economic development for strategy detailing. We additionally investigate the effect of financial, social, and political globalization on economic growth and the impact of imprisonment, disconnecting both monetary and social globalization. We expect that fossil fuel byproducts immediately affect financial development as filthy information (Bekun et al. 2019). All in all, we hope to affirm that rising fossil fuel byproducts will prompt climbing economic development (Cogollo et al. 2020) to decide the versatility connection between these factors and actuate the effect of the decrease in fossil fuel byproducts on economic growth in China during the 2020 repression.

**Hypothesis (H0).** *There is an unknown relationship between COVID-19 and the GDP of China.*

**Hypothesis (H1).** *There is a positive relationship between COVID-19 and the GDP of China.*

**Hypothesis (H2).** *There is a negative relationship between COVID-19 and the GDP of China.*

The present study analyzes the effect of COVID-19 on the economy of China, and it also studies whether economic and social activities in China, a highly industrialized nation, generate pollution emissions. For this purpose, many studies explain this relationship (Shittu et al. 2021).

**Hypothesis (H3).** *Economic and social isolation adversely affects Chinese economic growth due to the COVID-19 pandemic.*

The central hypothesis that describes the Sustainable Development Goals-3 of the health sector is that health is a primary necessity, and each country focuses on health prosperity. In the context of the global pandemic for the case of China, the present study seeks to understand the effect of social isolation and its implications on economic growth while considering health status.

Our primary model represents the effects of both economic and social globalization and is isolated. To understand the lockdown assumed by the Chinese administration during the COVID-19 crisis, a model from Zhang and Hu (2021) is followed.

For this purpose, our model is given below:

$GDP_t = \alpha_0 + \alpha_1 CO_2 t + \alpha_3 EG_t + \alpha_4 SG_t + \alpha_5 PG_t + \varepsilon_{it}$

GDP = gross domestic product

$CO_2$ = per capita carbon emission

EG = economic globalization

SG = social globalization

PG = political globalization

## 4. Results and Discussion

### 4.1. Statistical Analysis

The primary purpose of our study is to investigate the impact of COVID-19 on the Chinese economy. The table shows the descriptive statistical analysis of the variables which are chosen for the model.

Descriptive Statistics

In data analysis and estimation of the model, descriptive analysis always assists in understanding the data, giving the mean, median, mode, standard deviation minimum, and maximum values of variables. This analysis helps the researchers by depicting the range and essential characteristics of data. So, Table 2 presents the mean, median, standard deviation, minimum, and maximum values of all variables.

**Table 2.** Descriptive Statistics.

| Variable | Obs | Mean | Std. Dev | Min | Max |
|----------|-----|------|----------|-----|-----|
| GDP | 24 | 2.47 | 1.62 | −2.53 | 4.75 |
| SG | 24 | 60.16 | 23.09 | 15.16 | 98.18 |
| $CO_2$ | 24 | 1.63 | 0.78 | 0.46 | 3.40 |
| EG | 24 | 21.34 | 1.50 | 17.80 | 23.67 |
| PG | 24 | 21.02 | 1.36 | 18.38 | 23.14 |

Source: Authors calculations based on EViews 12.0.

Descriptive analysis is wholly explained in Table 2. Table 2 has 24 observations of each of the variables. GDP, SG, $CO_2$, and EG are variables. The dependent variable is GDP, whereas the independent variables are SG, $CO_2$, and EG. The mean value of GDP is

2.47 with a typical value of 1.62, and the mean value of social globalization is 60.16 with a standard deviation of 23.09. The mean value of $CO_2$ is 1.63 with a standard deviation of 0.78, and the mean value of economic globalization is 21.34 with a standard deviation of 1.50.

Table 2. Clearly depicts 24 observations, and the third column shows the central tendency of every variable. The standard deviation shows the dispersion by which data lies from its central value and t. The minimum and maximum values help determine the data ranges of variables used in the study.

### 4.2. Correlation Analysis

Correlation analysis helps to show the association among all variables, and this analysis gives relevant results about association among variables (Cohen 2013). Table 3 presents the correlation matrix that clears the relationship strength among all variables. Moreover, correlation analysis further assists in knowing whether variables are perfectly correlated or not. The nonexistence of perfect correlation considers well-organized data used in further analysis.

**Table 3.** Correlation Matrix.

|  | **GDP** | **SG** | **CO$_2$** | **EG** | **PG** |
|---|---|---|---|---|---|
| GDP | 1.00 |  |  |  |  |
| SG | 27 | 1.00 |  |  |  |
| CO2 | 27 | 0.12 | 1.00 |  |  |
| EG | 27 | −0.56 | 0.41 | 1.00 |  |
| PG | 27 | −0.63 | 0.39 | 0.96 | 1.00 |

Source: Author's calculations based on EViews 12.0.

Table 3 clearly suggests that all variables are associated with each other; however, there is not enough evidence about the presence of perfect multi-correlation. The value 1 shows the perfect correlation, but not a single variable is perfectly correlated. Therefore, we can trust this model and use it in regression analysis.

### 4.3. Unit Root Test

This section of the study utilizes unit root tests to identify the stationary level of the data set of a variable. The previous chapter detailed discussion about untaken unit root tests, and here we execute the unit root test. Different unit root tests are available, such as ADF (Augmented. Dickey-Fuller), PP (Phillips-Perron), Levin and Chu, and IPS unit root tests. For this study, ADF and PP tests of the unit root are performed to check the stationary of the data set. Accurate estimation of a dataset requires constant mean and variance that is independent of time; moreover, this situation leads to a stationary dataset.

Table 4 shows the results of the ADF test, and the unit root test is applied on variables separately. All variables are stationary at a level, so the order of integration is I(0), and we can suggest that there is no issue of a unit root. Moreover, some variables are significant at 2%, and the remaining are significant at a 5% or 10% level of significance. Foreign direct investment, GDP growth rate, and inflation rate are significant at a 2% level of significance. The other variables, including $CO_2$, EG, SG, and PG, are significant at a 5% level of significance. These results suggest that ordinary least square (OLS) is an accurate estimation method because variables are stationary at level.

**Table 4.** ADF Unit Root test.

| Variable | LEVEL | | First Difference | | Order of Integration |
|---|---|---|---|---|---|
| | Intercept | Trend and Intercept | Intercept | Trend and Intercept | |
| GDP | −3.30 * | −3.44 ** | −7.26 * | −7.18 ** | I(0) |
| SG | −0.45 | −1.93 ** | −5.57 * | −5.63 | I(0) |
| CO2 | −3.43 * | −3.35 ** | −5.16 * | −5.11 * | I(0) |
| EG | −2.57 | −2.93 ** | −4.09 * | −4.03 | I(0) |
| PG | −2.95 ** | −3.38 *** | −3.54 * | −3.50 ** | I(0) |

Source: Authors estimation based on EViews 12.0. H0: indicates the stationary dataset that is the null hypothesis (absence of unit root). Critical values are 10%, 5%, and 2%, and values of LM less than critical values indicate the acceptance of H0 (null hypothesis). This is again the case because the dataset has no unit root. * Significant at 10% level of significance. ** Significant at 5% level of significance. *** Significant at 1% level of significance.

Table 5 presents the PP (Phillips Perron) test results, and the unit root test is applied on variables separately. All variables are stationary at a level, so the order of integration is I(0), and we can suggest that there is no issue of a unit root. Moreover, some variables are significant at 2%, and the remaining are significant at a 5% level of significance. Foreign direct investment, GDP growth rate, and inflation rate are significant at a 2% level of significance. The other variables, external debt, gross fixed capital formation, gross capital formation, and unemployment rate, ($CO_2$, PG, SG, EG) are significant at a 5% level of significance. So, these results suggest that OLS (Ordinary least square) is an accurate estimation method because variables are stationary at level.

**Table 5.** PP Unit Root test.

| Variable | LEVEL | | First Difference | | Order of Integration |
|---|---|---|---|---|---|
| | Intercept | Trend and Intercept | Intercept | Trend and Intercept | |
| GDP | −3.32 * | −3.34 * | −8.24 * | −8.06 * | I(0) |
| SG | −0.35 | −1.93 ** | −5.58 * | −5.69 * | I(0) |
| $CO_2$ | −2.17 ** | −2.23 ** | −10.08 * | −5.11 * | I(0) |
| EG | −2.11 | −2.73 ** | −4.09 * | −4.03 | I(0) |
| PG | −3.21 ** | −3.67 ** | −3.74 * | −3.60 ** | I(0) |

Source: Author's estimation based on EViews 12.0. H0: indicates the stationary data set that is the null hypothesis (absence of unit root). Critical values are 10%, 5%, and 2%, and values of LM less than critical values indicate the acceptance of H0 (null hypothesis). This is again the case data set has no unit root. * Significant at 10% level of significance. ** Significant at 5% level of significance.

### 4.4. Estimation of Model

Table 6 represents the panel OLS regression results. The results indicate that economic growth and $CO_2$ have a positive relationship and are highly significant. It demonstrates that if $CO_2$ increases in China, economic growth also increases. The result indicates that if 1% increases in $CO_2$, then 3.69 units increase in the economic development of these countries. Economic globalization has a positive and significant impact on economic growth, and it indicates that if economic globalization increases in China, economic growth also increases. The result indicates that if a 1% increase occurs in economic globalization, then 3.40 units increase in economic growth. The results indicate that all the variables have positive and highly significant impacts on economic growth. The R square shows that these independent variables' change in economic growth is 87%.

**Table 6.** Estimation of Model.

| Dependent Variable GDP Growth | | | | |
|---|---|---|---|---|
| **Variable** | **Coefficient** | **Std. Err** | ***t* Value** | **Prob.** |
| SG | 0.030 | 0.2185 | 0.137 | 0.892 *** |
| $CO_2$ | 3.693 | 0.3696 | 9.991 | 0.000 * |
| EG | 3.40 | 0.426 | 7.98 | 0.000 * |
| PG | 0.243 | 0.134 | 1.813 | 0.084 ** |
| cons | 4.09 | 6.57 | 0.62 | 0.54 *** |
| Number of obs | | 24 | | |
| R-squared | | 0.87 | | |
| Adj R-squared | | 0.83 | | |

Source: Author's calculations based on EViews 12.0. * Result is significant at a 2% level of significance. ** Result is significant at a 5% level of significance. *** Result is significant at a 10% level of significance.

### 4.5. Diagnostic Tests

Table 7 explains the Wald test, which discusses the heteroscedasticity presence/absence in data. The F statistic value is 7.984160, the probability of F-statistic is 0.000, the chi-square value is 39.92080, and its possibility is 0.000.

**Table 7.** Wald Test.

| **Test Statistic** | **Value** | **df** | **Probability** |
|---|---|---|---|
| F-statistic | 7.984160 | (5, 127) | 0.0000 |
| Chi-square | 39.92080 | 5 | 0.0000 |

Author's calculation by EViews 9.5.

### 4.6. Breusch–Pagan Test for Heteroskedasticity

The presence of heteroskedasticity in a dataset can mislead the results, and we can rely on these results in policy-making or recommendations. Different tools and tests detect heteroskedasticity, but Breusch–Pagan is a significant test for the time series data set. Many previous studies used the same technique to detect the presence of heteroskedasticity, and among these studies (Halunga et al. 2017) can use as reference. The condition of Homoscedasticity indicates that error terms have the same or constant variance that is the fundamental property of CLRM. In applying this test, we have to make null and alternative hypotheses where the null hypothesis suggests there is no heteroskedasticity. On the other hand, Ha or the alternative hypothesis shows the presence of heteroskedasticity in the data set.

H0 = $\sigma1^2 = \sigma2^2 = \cdots = \sigma^2$
H0 = $\sigma1^2 \neq \sigma2^2 = \cdots \neq \sigma^2$
So:

Null hypotheses H0 = reschedules are not heteroscedastic, or there is no presence of heteroskedasticity

Alternative hypothesis Ha = the variance of error term does not contact or presence of heteroskedasticity.

Table 8 indicates that the prob. value of F statistics is 0.72 and prob values are 0.65, 0.57, and 0.91. Some values did not meet the minimum 5%, 2%, and 10% significance levels. Therefore, we have enough proof to accept the null hypothesis and conclude there is no presence of heteroskedasticity. The error term has a constant variance of data series, and we can use this dataset in other estimations.

**Table 8.** Breusch–Pagan test for heteroskedasticity.

| Breusch-Pagan Test | | | |
|---|---|---|---|
| F Statistics | 0.72 | Prob F | 0.65 |
| Obs R Squared | 5.70 | Prob Chi-square7 | 0.57 |
| Scaled Explained SS | 2.67 | Prob Chi-square7 | 0.91 |

Source: Author's calculations based on EViews 12.0.

## 5. Conclusions

The coronavirus pandemic that has ravaged China since December 2019 has put its economy to the test. China's GDP grew 5.94% in 2019 to $142.29.94 billion. China's economy shrank 6.8% in Q1 2020 due to nationwide lockdowns during the COVID-19 outbreak. It was the first decline since Beijing began reporting quarterly GDP in 1992. Then China's economy recovered from the COVID-19 pandemic. The National Statistics Bureau reported that China's gross domestic product increased 6.5% in the fourth quarter of 2020, exceeding the 6% growth rate in late 2019 before the Coronavirus took hold. China's economy grew by 2.3% in 2020, becoming the only major economy to do so during a year of global devastation. Like the United States, Europe, India, and Japan, other major nations and geopolitical competitors are battling a winter wave. In 2020, China's GDP will reach 100 trillion Yuan (15 trillion USD). With the pandemic over, innovation and digitization reignited China's economic growth. The economy grew 18.3% year-on-year in the first quarter of 2021. It's the most significant GDP increase since China began tracking quarterly GDP in 1992. China's GDP grew 7.9% in the second quarter and 2.3% in the third quarter compared to last year. China's economy is expected to grow by 5.5% in the fourth quarter and by 8.5% this year. As low base effects fade and the economy returns to its pre-COVID-19 trend growth, next year's growth is expected to slow to 5.4% (Tian 2021).

As a result, the stationarity of fossil fuel byproducts determines the approaches value. Our study also has limitations because it does not consider the impacts of specialized power use (Sanzo-Perez et al. 2017). In any case, the current study focuses on the GDP flexibilities of fossil fuel byproducts and how the globalization interaction is measured. If we only consider fossil fuel byproducts as a financial development result, our experimental results may be misleading (Liu 2021). The effects of monetary and social globalization are also considered. The absence of monetary and social globalization is due to the Chinese experts financial disengagement caused by the COVID-19 flare-up.

**Author Contributions:** Conceptualization, methodology, software, and validation: Z.H.; formal analysis: Z.H.; investigation: Z.H.; resources: Z.H.; data curation: Z.H.; writing—original draft preparation: Z.H.; writing—review and editing: Z.H.; visualization: H.H.; supervision: H.H.; project administration: M.A.M.; funding acquisition: M.A.M. All authors have read and agreed to the published version of the manuscript.

**Funding:** This research received no external funding.

**Institutional Review Board Statement:** Not applicable.

**Informed Consent Statement:** Not applicable.

**Data Availability Statement:** The data presented in this study are available on request from the corresponding authors.

**Acknowledgments:** The authors warmly thank friends and colleagues for their continuous support and advice over the years.

**Conflicts of Interest:** The authors declare no conflict of interest. The funders had no role in the design of the study, in the collection, analyses, or interpretation of data, in the writing of the manuscript, or in the decision to publish the results.

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
