# Peer review of "The Potential Impact of COVID-19 on the Chinese GDP, Trade, and Economy"

_economies, doi:10.3390/economies10040073_

Round 1

Reviewer 1 Report

The article on the impact of COVID-19 on the Chinese economy is a good study on this phenomenon. The method and data applied to this study are correct. I only recommend two revisions:

  • Mentioning some alternative methods that could be used in the investigation of the COVID-19 impact;
  • Adding the research question and the aim of the study in the introduction and abstract.

Author Response

, Introduction sections do not have subsections. I would recommend 1.1., 1.2., and 1.3. be moved down the paper into the Lit. Rev. or Methods sections.

Response: Yes, the introduction is improved and some subsections are added in the literature review sections.

  1. Lit. Rev. of studies on COVID

-lit. rev. – is not well-written. Rewrite it

-the idea is properly formulated, however, additional references are needed.

 Response: Yes, the literature review was rewritten.

  1. Source of Data

-this section should be labelled: Materials and Methods

-the formulae are sound and well-done.

-the hypotheses are also well thought out. The authors should be commended for this.

 Response: All the topics labeled as said.

  1. Results

-sufficient

The results are well done – overall. Compliments to the authors.

-this section should be renamed: Results and Discussion

 Response: We have revised the label as you said.

  1. Conclusion

-not sufficient, it should be rewritten

-The authors should highlight the current limitations of their study, and briefly mention some precise directions (future ideas) for the work.

 Response: We have rewritten the Conclusion section.

Finally, ENGLISH: The paper was quite difficult to read and will require a FULL ENGLISH LANGUAGE overhaul. It should be checked for English typographical mistakes, e.g., when quoting authors in a sentence there are no brackets—Line 262 should be:

Liu and Hu (2020) explored the short-run and […]

Response: We revised please see the new version.

Huge parts of the paper also do not read well. On English alone the paper could be rejected; however, with a native English read through the paper’s language can be properly improved.

Response: We have invited experts in the field to review and revise our manuscript.

At length, in general, I would like to thank the authors for submitting a highly needed and timely paper on COVID in China relating to (domestic) economic factors. After the revisions, the paper may look quite different. It is my recommendation that the authors complete the steps and comments before the paper is considered publishable.

Response: thank you for your positive comments.

Reviewer 2 Report

Manuscript ID: economies-1549936

Article Title: The Potential impact of covid-19 on the Chinese GDP, trade, and Economy

Review Report

Recommendation: I would recommend this article for publication pending REVISIONS. The paper is very interesting and warrants publication.

The paper is consistent with MDPI ECONOMIES and fits in with the overall journal scope. More specifically, it was submitted to the Special Issue “The Impact of COVID-19 on Financial Markets and the Real Economy” which it also fits in very well with.

The paper looks at COVID-19 in terms of the dramatic affects to the financial market, economy, and society within China. It examines a series of impact related issues, specifically, China’s policy responses to this shock—fiscal and institutional measures. The study explores future concerns and warrants publication. The research findings indicate that the occurrence of coronavirus in China has significant impacts across the globe. Due to this virus threat, the authors state that non-Chinese fear mixing up with the Chinese population. The use of tourism is used as an example. The rapid spread of the virus in China has stopped the average life of its citizenry. Finally, the authors state that the intensification of the coronavirus is being thought to have a long-term impact on China’s economy. The authors claim need to be properly justified with references. The policy recommendations need to be also explained clearly for the right path of recovery within the shortest possible time.

First, the lit. rev. and references are mostly sufficient, however, additional referencing is needed in the introductory sections, especially subsections 1.1. and 1.2, and throughout the paper, including sections 2 and 3. The research idea is good and its significance novel and up to date; as it stands I would consider the paper’s novelty as medium. The paper length is fine. The paper is NOT written clearly though—it will be necessary for a proper read through of the work for English. The statistical work and model specification and hypotheses development is sound; the results are also clear. There are a lot of valuable studies in the scientific literature related to the subject of the manuscript to which the authors can compare to. This comparison will highlight even more the novel aspects that their COVID research by bringing out and contrasting that with existing studies. As such, additional references are needed.

The paper is relevant and interesting and shows promise. Sufficient background into the topic has been presented.

       Abstract

-not sufficient, should be rewritten

  1. Introduction

-not sufficient

-the authors have written a sufficient first part to the introduction that highlights the topic and properly sets the paper’s theoretical framing. I would encourage the authors to add additional references to better set the stage for the paper and work they are presenting.

-also, traditionally, Introduction sections do not have subsections. I would recommend 1.1., 1.2., and 1.3. be moved down the paper into the Lit. Rev. or Methods sections.

  1. Lit. Rev. of studies on COVID

-lit. rev. – is not well-written. Rewrite it

-the idea is properly formulated, however, additional references are needed.

  1. Source of Data

-this section should be labelled: Materials and Methods

-the formulae are sound and well-done.

-the hypotheses are also well thought out. The authors should be commended for this.

  1. Results

-sufficient

The results are well done – overall. Compliments to the authors.

-this section should be renamed: Results and Discussion

  1. Conclusion

-not sufficient, it should be rewritten

-The authors should highlight current limitations of their study, and briefly mention some precise directions (future ideas) for the work.

Finally, ENGLISH: The paper was quite difficult to read and will require a FULL ENGLISH LANGUAGE overhaul. It should be checked for English typographical mistakes, e.g., when quoting authors in a sentence there are no brackets—Line 262 should be:

Liu and Hu (2020) explored the short-run and […]

Huge parts of the paper also do not read well. On English alone the paper could be rejected; however, with a native English read through the paper’s language can be properly improved.

At length, in general, I would like to thank the authors for submitting a highly needed and timely paper on COVID in China relating to (domestic) economic factors. After the revisions, the paper may look quite different. It is my recommendation that the authors complete the steps and comments before the paper is considered publishable.

Author Response

Introduction sections do not have subsections. I would recommend 1.1., 1.2., and 1.3. be moved down the paper into the Lit. Rev. or Methods sections.

Response: Yes, the introduction is improved and some subsections are added in the literature review sections.

  1. Lit. Rev. of studies on COVID

-lit. rev. – is not well-written. Rewrite it

-the idea is properly formulated, however, additional references are needed.

Response: Yes, the literature review was rewritten.

  1. Source of Data

-this section should be labelled: Materials and Methods

-the formulae are sound and well-done.

-the hypotheses are also well thought out. The authors should be commended for this.

Response: All the topics labeled as said.

  1. Results

-sufficient

The results are well done – overall. Compliments to the authors.

-this section should be renamed: Results and Discussion

Response: We have revised the label as you said.

  1. Conclusion

-not sufficient, it should be rewritten

-The authors should highlight the current limitations of their study, and briefly mention some precise directions (future ideas) for the work.

Response: We have rewritten the Conclusion section.

Finally, ENGLISH: The paper was quite difficult to read and will require a FULL ENGLISH LANGUAGE overhaul. It should be checked for English typographical mistakes, e.g., when quoting authors in a sentence there are no brackets—Line 262 should be:

Liu and Hu (2020) explored the short-run and […]

Response: We revised please see the new version.

Huge parts of the paper also do not read well. On English alone the paper could be rejected; however, with a native English read through the paper’s language can be properly improved.

Response: We have invited experts in the field to review and revise our manuscript.

At length, in general, I would like to thank the authors for submitting a highly needed and timely paper on COVID in China relating to (domestic) economic factors. After the revisions, the paper may look quite different. It is my recommendation that the authors complete the steps and comments before the paper is considered publishable.

Response: thank you for your positive comments.